# Challenges Related to Acquisition of Physiological Data for Physiologically Based Pharmacokinetic (PBPK) Models in Postpartum, Lactating Women and Breastfed Infants—A Contribution from the ConcePTION Project

**DOI:** 10.3390/pharmaceutics15112618

**Published:** 2023-11-12

**Authors:** Martje Van Neste, Annick Bogaerts, Nina Nauwelaerts, Julia Macente, Anne Smits, Pieter Annaert, Karel Allegaert

**Affiliations:** 1Clinical Pharmacology and Pharmacotherapy, Department of Pharmaceutical and Pharmacological Sciences, KU Leuven, 3000 Leuven, Belgium; karel.allegaert@uzleuven.be; 2L-C&Y, KU Leuven Child & Youth Institute, 3000 Leuven, Belgium; annick.bogaerts@kuleuven.be (A.B.); anne.smits@uzleuven.be (A.S.); 3Department of Development and Regeneration, KU Leuven, 3000 Leuven, Belgium; 4Faculty of Health, University of Plymouth, Devon PL4 8AA, UK; 5Drug Delivery and Disposition, Department of Pharmaceutical and Pharmacological Sciences, KU Leuven, 3000 Leuven, Belgium; nina.nauwelaerts@kuleuven.be (N.N.); julia.macente@kuleuven.be (J.M.); pieter.annaert@kuleuven.be (P.A.); 6Neonatal Intensive Care Unit, University Hospitals Leuven, 3000 Leuven, Belgium; 7BioNotus GCV, 2845 Niel, Belgium; 8Department of Hospital Pharmacy, Erasmus University Medical Center, 3000 CA Rotterdam, The Netherlands

**Keywords:** pharmacokinetics, physiologically based pharmacokinetic (PBPK) modelling and simulation, pregnancy, lactation, breastfeeding, infant, pharmacotherapy

## Abstract

Physiologically based pharmacokinetic (PBPK) modelling is a bottom-up approach to predict pharmacokinetics in specific populations based on population-specific and medicine-specific data. Using an illustrative approach, this review aims to highlight the challenges of incorporating physiological data to develop postpartum, lactating women and breastfed infant PBPK models. For instance, most women retain pregnancy weight during the postpartum period, especially after excessive gestational weight gain, while breastfeeding might be associated with lower postpartum weight retention and long-term weight control. Based on a structured search, an equation for human milk intake reported the maximum intake of 153 mL/kg/day in exclusively breastfed infants at 20 days, which correlates with a high risk for medicine reactions at 2–4 weeks in breastfed infants. Furthermore, the changing composition of human milk and its enzymatic activities could affect pharmacokinetics in breastfed infants. Growth in breastfed infants is slower and gastric emptying faster than in formula-fed infants, while a slower maturation of specific metabolizing enzymes in breastfed infants has been described. The currently available PBPK models for these populations lack structured systematic acquisition of population-specific data. Future directions include systematic searches to fully identify physiological data. Following data integration as mathematical equations, this holds the promise to improve postpartum, lactation and infant PBPK models.

## 1. Introduction

The European Medicines Agency defines a physiologically based pharmacokinetic (PBPK) model as “a mathematical model that simulates the concentration of a drug over time in tissue(s) and blood, by taking into account the rate of the drug’s absorption into the body, distribution in tissues, or metabolism and excretion (ADME) on the basis of interplay between physiological, physicochemical and biochemical determinants” [1]. In other words, PBPK modelling allows a bottom-up prediction of pharmacokinetics based on the integration of population-specific data regarding physiology with medicine-specific data regarding disposition.

PBPK modelling has already been embraced by the pharmaceutical industry and regulatory agencies, e.g., to predict drug–drug or drug–food interactions, or to support an initial dose finding in first-in-human and paediatric trials [1,2,3]. In addition, properly applied PBPK models have the potential to reduce the number of subjects in early phase clinical trials and the time and resources required in paediatric medicine development [4].

Despite the evolutions in physiologically based modelling and expanding insights into pharmacokinetics, there is still limited, poorly standardized and scattered knowledge regarding the physiology of special populations. These special population models require integration of underlying (patho)physiological data, collected through systematic searches and observed data, and subsequently converted to mathematical equations to develop and evaluate these tools [2]. Using structured integration, the predictive performance of PBPK modelling in special adult populations (like hepatic dysfunction, chronic kidney disease (CKD) or pregnancy and lactation) improved [2,5,6]. Along the same line, neonatal (and preterm) PBPK models were developed, but with still insufficient reflections on the specific physiology of breastfed versus formula-fed infants [7,8]. To fully operationalize the potentials of postpartum, lactation and breastfed infant PBPK models and generate more confidence in their applications, further collection of physiological data based on systematic review approaches specific in these populations is warranted [2,9].

The recommended standard for infant feeding, according to the World Health Organization (WHO), is exclusive breastfeeding up to 6 months of age and partial breastfeeding from 6 months of age, supported by the benefits described for both mother and infant [10]. Maternal health benefits include a reduced risk to develop ovarian and breast cancer and type II diabetes and a higher recovery of pre-pregnancy weight [11,12]. Advantages for the child’s health relate to a lower incidence of infections, protection against later metabolic disorders and a better performance on intelligence tests [13,14]. However, the majority of women require pharmacotherapy during the postpartum period and this number is rising due to later-age pregnancies and an increased prevalence of chronic pathologies [15]. Similarly, the incidence of gestational diabetes mellitus (GDM) and obesity in women of a reproductive age is increasing, which further results in a higher risk for associated perinatal and postpartum complications [16,17].

Medicine-in-milk safety assessments of most medicines are lacking, which results in an extensive information gap regarding medicine use and safety during pregnancy and lactation [18]. These uncertainties result in the cessation of breastfeeding or in poor adherence to pharmacotherapy in mothers, when medicine exposure in infants through human milk would generally be minimal [14,18]. PBPK modelling is an in silico methodology very likely to improve this setting. This article highlights the challenges of developing PBPK models during postpartum, lactation and the first year of life with a focus on the changes in the population-specific physiology. PBPK models can improve exposure prediction of medicines in human milk and can contribute to removing the above-mentioned uncertainties, by structurally incorporating physiological data of postpartum and breastfeeding women and infants. Knowledge on these physiological changes is scattered, limited and non-standardized and needs systematic research.

The aim of this review is to illustrate and emphasize the limited and fragmented knowledge on the population-specific physiological changes during postpartum, lactation and infancy. The intention is to highlight the need and challenges to collect these physiological data in a structured and systematic way from the literature and to subsequently convert these data into specific mathematical functions. We will first provide relevant illustrations of the specific physiology of postpartum, human milk, and breastfed infants, respectively. This will be followed by a discussion on the possible search approaches to collect these findings in updated PBPK models.

## 2. Postpartum Maternal Weight Retention

Weight and body composition are crucial covariates of pharmacokinetics. Maternal weight changes during pregnancy and postpartum are therefore relevant to be incorporated as specific equations in PBPK models of these populations.

Around 30% of gestational weight gain (GWG) can be accounted for by the growth of the foetus, placenta and amniotic fluid. This suggests that maternal factors, such as the increased circulating volume and fat mass, cause the majority of weight gain during pregnancy [19]. Guidelines for ranges of GWG were published by the Institute of Medicine (IOM), based on pre-pregnancy BMI categories, to prevent postpartum weight retention (PPWR) and the consequences in mothers. Underweight women (BMI < 18.5 kg/m^2^) are recommended to gain 12.5 to 18 kg during their pregnancy, normal weight women (BMI 18.5–24.9 kg/m^2^) 11.5 to 16 kg, overweight women (BMI 25.0–29.9 kg/m^2^) 7 to 11.5 kg and obese women (BMI ≥ 30.0 kg/m^2^) 5 to 9 kg [20]. A technical advisory group has recently been established to develop new GWG recommendations [21].

Still, pregnancy is a risk factor for increasing body weight and developing obesity in women of a reproductive age because of their GWG and PPWR [22]. Furthermore, obesity has become a worldwide issue as approximately 50% of women of reproductive age are overweight or obese [19]. This worldwide increasing pre-pregnancy weight further adds to the postpartum maternal weight patterns and its variability and effect on pharmacokinetics.

Multiple studies described excessive GWG in approximately a third of pregnant women and an overall increase in GWG in several industrialized countries. A systematic review on excessive GWG reported GWG of 19 kg in normal weight women (3 kg above the IOM recommendations), 16 kg in women with overweight (4.5 kg above) and 14 kg in women with obesity (5 kg above) [22]. Excessive GWG can lead to multiple maternal and neonatal complications, such as an increased risk of caesarean section and foetal macrosomia and an increased risk for high PPWR [19,23,24].

Failure to lose this GWG is a risk factor for obesity and associated morbidities later in life, such as type II diabetes mellitus and cardiovascular diseases. Unfortunately, the majority of women fail to lose their pregnancy weight with a mean weight retention of 4.5 kg 12 months postpartum [22]. Furthermore, and next to GWG, increases in body fat percentage and waist circumference have been described from 12 weeks of gestation (29 ± 0.5% and 78 ± 1 cm, respectively) to 6 weeks postpartum (31 ± 0.5% and 81 ± 1 cm, respectively), with a decrease to 6 months after childbirth (30 ± 1% and 80 ± 1 cm, respectively) [25]. These postpartum maternal weight patterns, their variability and covariates involved are also relevant to inform postpartum maternal PBPK models.

In women with excessive GWG, a faster weight loss during the first 6 weeks postpartum has been described compared to women with GWG within and below the guidelines; although, the weight retention thereafter was higher compared to women with GWG within the guidelines (5 kg vs. 2 kg) [19]. Similarly, overweight and obese women showed a faster weight loss from 2 weeks postpartum compared to normal weight women. At 6 weeks postpartum, women with obesity showed a lower mean weight retention compared to normal weight women (2 kg vs. 6 kg) but had a higher fat percentage (39% vs. 31%) and a higher waist circumference (98 cm vs. 82 cm) [19,22]. From 6 weeks to 6 months after delivery, postpartum weight loss was slower in overweight women compared to normal weight women with a stagnation of PPWR in overweight women after 6 months. A slight increase in PPWR was seen in women with obesity between 6 weeks and 6 months postpartum, with a small monthly increase thereafter [22]. In terms of long-term maternal weight retention, excessive GWG in women leads to PPWR of 3.1 kg and 4.7 kg after 3 years and 15 years postpartum, respectively [23]. Excessive GWG and PPWR may result in a cycle of BMI increases that can be repeated in future pregnancies [24]. Around 50% of women with excessive GWG have a higher BMI during their next pregnancy compared to their pre-pregnancy BMI of the previous pregnancy. This PPWR means that they have an increased risk of pregnancy and birth-related complications in the next pregnancy, including pregnancy induced hypertension and gestational diabetes [19].

The incidence of gestational diabetes mellitus (GDM) is rising, together with the worldwide increase in obesity [17]. Maternal and foetal outcomes are influenced by GDM, with an increased risk of complications during pregnancy, perinatal period and postpartum, such as pre-eclampsia, preterm delivery and foetal macrosomia. In 70–85% of patients diagnosed with GDM, the management is sufficient with physical activity, lifestyle modifications and GWG management. However, up to 15–30% of patients require pharmacotherapy, such as insulin and oral hypoglycaemic agents. In addition, women with GDM have a 10-fold risk of developing type II diabetes mellitus in the future, have a higher risk of being overweight and developing metabolic syndrome. Therefore, mothers are recommended to exercise to prevent GDM before the pregnancy or early in the pregnancy and women who experienced GDM should be monitored for the development of type II diabetes mellitus [17,26].

Highly relevant to lactation-related PBPK model construction, PPWR is also inversely associated with breastfeeding, since it requires more energy (roughly 500 kcal/day) and possible fat mobilization [27,28]. Weight loss after 3 and 6 months postpartum in exclusively breastfeeding women was 0.7 and 0.5 kg/month, respectively, which resulted in losing more than 85% of the gestational weight gain [29]. Longitudinal data found that breastfeeding, and more specifically exclusive breastfeeding, is connected to a lower weight gain for 8 to 10 years postpartum and might be associated with long-term weight control [27]. It is reported that the initiation of exclusive breastfeeding and duration of breastfeeding, in particular 3 to 6 months, is required to show an influence on maternal weight and that little influence is seen when mothers breastfeed for more than 6 months. However, the evidence on the effects of lactation on maternal weight is inconsistent and conflicting [19,22,28]. In addition, as many women are questioning the energy restriction during breastfeeding for fear of decreased milk production, women often retain or increase their postpartum weight while breastfeeding [28].

To better explore the variability in exposure in different postpartum subpopulations, systematic searches on the specific characteristics of these populations to develop accurate mathematical functions should be used as an approach. This is of potential relevance for both maternal, as well as lactation-related exposure in infants.

## 3. Human Milk Intake and Composition

Daily human milk volume and composition evolve during the lactation period as well as during a single feeding moment.

Many approaches have been described to measure human milk intake in infants and to subsequently calculate infant medicine exposure through breastfeeding. Test-weighing and deuterium oxide dose-to-the-mother technique are the most often used to measure milk intake. The first method is the conventional technique to estimate milk intake by weighing the infant before and after each feeding. This procedure is used during the first weeks of life and is easy and direct with minimal interference with the lactation process, but it is time-consuming since the observer has to be present for up to 48 h. Studies found that this method is accurate but imprecise, as it underestimates or overestimates the human milk intake of a feeding by up to 15 mL in 95% of cases, which means that this is an unreliable method to be implemented in clinical practices. An alternative method for human milk intake estimation in clinical practice could be test-weighing after every feed over a 24 h period; however, these findings were not validated [30,31].

The deuterium oxide dose-to-the mother technique, on the other hand, is a non-invasive method that uses a stable, non-radioactive isotope to assess the human milk intake. The deuterium oxide is orally consumed by the mother, whereafter the disappearance in the mother’s and the appearance in the child’s saliva are monitored. The consumed amount of deuterium (0.1%) in these studies is far under the toxicity threshold (15%) and is stated to be non-toxic for mother and infant, since no adverse effects in mother or infant have been described. One of the benefits of this method is the possibility to measure the human milk intake in not only exclusively breastfed (EBF) but also partially breastfed (PBF) infants. However, this technique is fairly slow and measures the volume of human milk intake over a period of 14 days [32,33,34].

Yeung et al. recently quantified the daily human milk intake from birth to 1 year of life for input in term and preterm PBPK models [14]. Following a comprehensive search on terms related to premature and term infants, breastfeeding and volume, a regression equation for the weight-normalized human milk intake (WHMI), consistent with the observed data and similar to already existing linear regression models, has been described in term infants up to 6 months of postnatal age:$$WHMI\;\left(\mathrm{mL}/\mathrm{kg}/\mathrm{day}\right)=160.39\times \frac{0.232}{0.232-0.00252}\times \left(e^{-0.00252t}-e^{-0.232t}\right),$$
where *t* = days. The maximum WHMI for infants, exclusively breastfed up to 6 months of age, was set at 153 mL/kg/day at day 20, with a weighted mean feeding frequency of 7.7 feeds/day. This equation is similar to the human milk intake of breastfed preterm infants [14].

The peak volume of intake during the first month indicates that the greatest risk of medicine exposure in breastfed infants occurs at 2–4 weeks. The lower medicine metabolism and excretion in infants, together with the high medicine dose relative to their weight explains the risk for higher medicine exposure and toxic effects during the first months, especially in the preterm population [13,32]. Furthermore, infants are more likely breastfed and mainly exclusively breastfed during the first months of life, which might result in higher medicine exposure in younger infants. These findings are consistent with a review on adverse medicine reactions in breastfed infants, as almost 80% of cases of adverse reactions in infants were observed in the first 2 months, with approximately two-thirds of the cases in the first month of life [35]. Moreover, early postpartum is a specific lactation interval of interest as it covers both the shift from colostrum to mature milk, as well as a fast progressive increase in the daily milk intake (mL/kg/day), an important determinant of compound dosing through human milk (mg/mL × mL).

The daily frequency of feeds stays rather stable with the age of the EBF term infant, with no difference between boys and girls [36]. Oras et al. described that the feeding frequency in EBF preterm infants is higher as described in the term population; however, large variations are seen in the frequency of breastfeeding sessions [36,37]. Strong differences in breastfeeding frequency can be explained by biological and cultural perspectives, considering fewer feeding frequencies are described in studies in urban communities or developed countries, whereas higher frequencies are reported in studies in rural communities or in developing countries [14,37]. However, it is suggested that the feeding frequency has a minimal impact on medicine exposure in breastfed infants [38].

Furthermore, fresh human milk holds enzymatic activities, and these enzymes may also affect the medicine absorption or enterohepatic recirculation patterns [39,40]. Breastfed infants have an increased enterohepatic recirculation and a decreased bilirubin clearance, as human milk has considerably more ß-glucuronidase activity than formula milk (419 units/mL vs. 6 units/mL). This difference in ß-glucuronidase activity is more pronounced in the first days postpartum, with a decrease of almost 70% in the third week [39]. Similarly, amylase levels are highest in the initial phase but remain constant in mature human milk up to 27 months postpartum [41]. Conversely, the levels of bile salt-stimulated lipase (BSSL) activity in human milk increases from colostrum to the third week of lactation (30.8 ± 1.09 U/mL vs. 42.6 ± 1.03 U/mL, respectively) and decreases after 6 weeks to levels similar to those in colostrum at 12 weeks (30.8 ± 1.23 U/mL) [42].

Finally, pH values, protein and fat content change in human milk over time, which can have an effect on medicine kinetics in milk [43,44]. The pH values of milk are decreasing from colostrum to 14 days postpartum (pH 7.45 vs. pH 7.04, respectively) but are rising again after 90 days with pH values of 7.40 at 300 days postpartum [44]. Human milk comprises more than 400 different proteins with high levels of whey protein in colostrum. The curve of protein content in human milk decreases during lactation and flattens after 7 months. Overall, mothers who delivered at full term express lower protein levels in their human milk than mothers who delivered preterm [43,45]. Human milk fat concentration is highly variable and increases with a longer breastfeeding duration and increases during each breastfeeding moment [46]. The initial milk of a feed, the foremilk, can contain less than half of the milk fat concentration of the hindmilk, the last milk of the feed [43]. Moreover, a diurnal variation has been described with a higher milk fat concentration during the evening than the morning (43 ± 9 g/L vs. 37 ± 10 g/L, respectively) [36].

Systematic searches to develop specific mathematical functions are needed to adequately explore the variability in human milk volume and composition.

## 4. The Specific Physiology of Breastfed Infants

The maturation of the pharmacokinetic processes in early infancy is driven by age or growth as the main covariates. This time-dependent physiology is further affected by non-maturational covariates like genetics or environmental factors (e.g., drug–food interaction, drug–drug interaction, drug-treatment modalities, disease) [47]. It is important to realize that lactation is also one of these environmental factors, as will be illustrated for gastric emptying, growth and body composition, and differences in medicine metabolism.

Gastric emptying is a clear example of the merged effects of maturation and diet. In (pre)term neonates and young infants, gastric emptying is slower, as evidenced by paracetamol absorption patterns [48]. However, in a meta-analysis on reported data from preterm neonates through to adults, the diet (aqueous > human milk > formula milk > semi-solid > solid) was the main driver of the gastric emptying, with a correlation between meal type (milk vs. (semi-)solid) and age. The mean gastric residence time (min) for human milk was 11% shorter compared to formula, with a faster gastric emptying (+15%) for an extensive hydrolysed formula compared to a partially hydrolysed or intact protein formula [49,50]. In the subsequently developed PBPK model for paediatric oral medicine absorption, the diet related parameter was simplified to either fasted, liquid fed or (semi)solid at 0.75, 1.18 and 1.8 h, respectively [51].

Infants fed human milk also display different patterns of growth, weight and body composition when compared to formula-fed infants. The average weight loss in the first days of life of breastfed infants compared to formula-fed infants has been inconsistent in the literature, presumably as a consequence of different feeding practices and protocols [52]. In healthy full-term breastfed infants, a part of this weight loss can be explained by a higher proportional loss in fat mass than fat-free mass [53]. Nevertheless, the time to birth weight recovery takes longer in breastfed than formula-fed infants (8.3 vs. 6.5 days) [54]. This pattern and its covariates have been integrated in a turnover model on physiological weight changes (weight loss and gain, with both varying rates over postnatal age) over the first week of life. Interestingly, this turnover model has been developed for either formula- or human-milk-fed infants, with main differences in the second part of the first week of life, so after the weight nadir (lowest postnatal weight) was reached (day 2–3), with a delayed regain of the birth weight in infants fed with human milk [55].

Growth charts for infants under 24 months with the healthy breastfed infant as standard (predominantly breastfed for at least 4 months), have been published by the WHO. Compared to growth charts based on more heterogeneous cohorts like the Centers for Disease Control and Prevention (CDC) reference cohort, slower growth and weight gain among breastfed infants between 3 and 18 months is normal (with a difference for length of about 0.5 SD between both charts) [56]. Consequently, for human milk related PBPK modelling efforts, the WHO growth charts are the best option.

Along the same line, maturational changes in body composition are also influenced by lactation. Exclusively breastfed infants display an increase in proportional fat mass from 8.8% at birth to 28.1% at 4 months (%FM was not different between both sexes at any study point) [57]. In a more recent meta-analysis on the impact of the method of feeding on body composition, one study reported that exclusively breastfed infants at 3 months of age had a higher fat mass (1.98 vs. 1.80 kg, +10%, *p* < 0.02) and a lower fat-free mass (4.15 vs. 4.32 kg, −4%, *p* < 0.01) compared to exclusively formula-fed infants. At 6 months, fat mass values remained significantly higher in exclusively breastfed boys (2.56 vs. 2.04 kg, +25%, *p* < 0.05) compared to formula-fed boys. However, observations were not consistent as the three other studies retained in the meta-analysis could not find these differences [58].

In addition to the age-related trends, studies also reported on differences in serum biochemistry between breastfed and formula-fed infants, with higher albumin (40 vs. 38 g/L; *p* = 0.04), liver enzymes (AST: 53 vs. 29 U/I; *p* = 0.04), bilirubin (1.1 vs. 0.4 mg/dL; *p* = 0.02) and triglycerides (135 vs. 101 mg/dL; *p* < 0.05), and lower urea nitrogen values (6.00 vs. 8.79 mg/dL; *p* < 0.05) in breastfed infants [59,60].

Finally, the relevance of human milk co-exposure with medicines or exogenous compounds in PBPK models tailored to breastfed infants are also reflected by in vivo observations on the effect of diet (lactation vs. formula) on medicine metabolism (e.g., caffeine, dextromethorphan) by the cytochrome P450 (CYP450) enzyme superfamily. Overall, the caffeine elimination rate constant (k_e_) was low at 2 weeks, with a subsequent increase over the first 6 months of life. This increase was significantly faster in formula-fed than in breastfed infants (k_e-formula_ = 0.009 e^0.0917x^ vs. k_e-human milk_ = 0.009 e^0.0676x^, with x = postnatal weeks). The caffeine example reflects the faster CYP1A2 maturation in formula-fed infants. A similar pattern, with a faster maturation, was also observed for dextromethorphan by the urinary molar 3-hydroxymorphinan/dextromethrophan ratio, reflecting maturational changes in CYP3A4 activity [61]. The impact of the type of feeding on maturational medicine metabolism for both probes used (caffeine, dextromethorphan) is further illustrated in Table 1. These findings confirm the impact of formula versus human milk feeding on the elimination half-life of caffeine in preterm infants (30–60 weeks postmenstrual age) as previously reported, with clearly different maturational trend lines with faster maturation in formula-fed infants, as seen in Figure 1 [62].

Claimed or suggested mechanisms to explain these different maturational patterns may relate to differences in intestinal microbial flora or the presence of enzyme inducers in formula (e.g., aryl hydrocarbon receptor ligands). Irrespective of these mechanisms, the phenotypic data strongly suggest that diet (human milk vs. formula) modulates medicine biotransformation up to the level of clinical relevance within the framework of human-medicine-related medicine exposure in lactating infants [61,63].

## 5. Discussion

There are still many challenges and opportunities when developing PBPK models tailored to the postpartum, lactating, and infant population, such as collecting, standardizing and incorporating the fast-changing maturational physiology changes of these populations. Developing PBPK models for these populations also depends on the PBPK software but are mainly developed from an existing and verified PBPK model for an adult, non-lactating population. Subsequently, this model is extended to resemble the medicine concentration profile in the targeted special population by implementing modifications in terms of characteristic physiology [64].

A currently available postpartum PBPK model is an extension of a previously developed pregnancy model with a compartment for breast tissue and for endometrium and myometrium of the uterus among other things, while pregnancy-specific compartments (e.g., foetus, placenta) are discarded [65]. Lactation PBPK models are developed from a verified adult, non-lactating population and extended with specific physiological data, such as a breast and human milk compartment [9,64,66]. The infant PBPK models are extrapolated from the adult PBPK model to represent the population, with physiological data irrespective of the type of feeding [67]. To add these extensions to the adult model, physiological data of the populations were searched for in the literature and were implemented as mathematical equations fitted to the observed data. However, many physiological data were included that were based on assumptions due to the limited data and non-systematic searches of the literature [64,65,67]. This means that data available in the literature might be missed when developing a postpartum, lactation and breastfed infant PBPK model.

Systematic searches to collect population-specific data carry the promise to support the development of PBPK models that adequately reflect the variability in physiology (and therefore drug exposure) in postpartum, lactating women and their breastfed infants. Data standardisation will be a key element in such a next-generation data collection expedition, to minimize the bias due to methodological differences between studies. For example, efforts should be made to develop a structured method for collecting clinical study data and processing the resulting physiological data with minimal inter-study variability. In this way, the impact of methodological heterogeneity between studies remains limited, while compilation of data in a meaningless summary can be avoided. The heterogeneity between clinical studies can be addressed using the meta-regression or subgroup analysis approach [68].

Based on the information and challenges mentioned in previous sections, it is clear that the current structured, but non-systematic search approaches used to generate PBPK models for these populations are suboptimal. Furthermore, additional data on physiology are needed. As a tool to facilitate structured collection and access to data, to monitor data acquisition and provide access to maternal and paediatric pharmacokinetics research and to facilitate incorporation of these data in modelling efforts, the National Institute of Child Health and Human Development (NICHD) established the Maternal and Pediatric Precision in Therapeutics (MPRINT) Hub [69].

In the currently available postpartum women population for PBPK models, all physiological data were implemented in order to return to their pre-pregnancy levels. For instance, if the mathematical equation based on collected data from the literature did not reach the pre-pregnancy levels in certain PBPK frameworks, an artificial value was added at least 14 weeks postpartum to force the data to their pre-pregnancy levels [65]. However, most mothers do not reach their pre-pregnancy weight in the first year postpartum due to PPWR, and pre-pregnancy levels were often not reached after 3 years in women with excessive GWG [22,23]. In addition, breastfeeding may have a favourable effect on PPWR and might be associated with long-term weight control, although inconclusive evidence was found [19,22,27,28]. Mathematical equations based on systematic searches of the literature for postpartum body weight evolution in PBPK platforms, differentiating between breastfeeding and formula-feeding mothers, can improve the PBPK predictive performance during the postpartum period, or could at least facilitate the exploration of different scenarios. At least, the potential impact of a further optimized postpartum model should be assessed including but not limited to weight optimization, in particular regarding the effect of breastfeeding on PPWR and body composition on the pharmacokinetics, e.g., the distribution of medicines during the postpartum period.

Many physiological data change during postpartum, including enzyme activity. For example, paracetamol metabolic clearance by glucuronidation via UDP-glucuronosyltransferase (UGT) enzyme activity is affected by oestradiol levels. This results in a lower paracetamol clearance during early postpartum in lactating women (as their oestradiol levels are lower) [70]. This indicates the need to further explore and incorporate postpartum clearance of medicines and more specifically, endocrine driven enzyme activity. Furthermore, there are some characteristic pathophysiological traits during the postpartum period, that can have an effect on pharmacokinetics, which are often not yet described in postpartum PBPK models. For example, albumin concentrations were not properly predicted by a postpartum PBPK model in women with Human Immunodeficiency Virus (HIV) [71].

Human milk intake, normalized for weight, of breastfed term and preterm infants up to 6 months old was fit to a mathematical equation, consistent with previous regression models. However, this equation was based on a comprehensive literature search instead of a more thorough systematic search, while no differentiation was made between the test-weighing and deuterium oxide dose-to-the-mother technique as a human milk intake measuring approach [14]. Nonetheless, to calculate infant medicine intake, the daily human milk intake is currently often assumed to be 150 mL/kg body weight, while it is recommended to additionally estimate the infant risk on 200 mL/kg/day as a worst-case scenario [72]. These worst-case scenarios should be considered in lactation-related PBPK models predicting medicine exposure in infants.

Colostrum is produced in low-volume quantities during early postpartum, which results in a minimal risk of medicine exposure (in absolute amounts) in the infant, and has a considerable immunologic function as well as a nutritional function [43]. While the composition of milk changes during the postpartum period, the enzymatic activities of fresh human milk change over time. The activity of ß-glucuronidase, amylase and BSSL, together with the composition of human milk, can have an effect on medicine absorption and clearance in the breastfed infant. Therefore, these physiological time-varying data and their impact on pre-absorption and absorption should be accounted for when developing a breastfed infant PBPK model.

Breastfed infants have a different maturational physiology compared to formula-fed infants, such as gastric emptying time, with an influence on medicine absorption. However, data on drug absorption in infants, breastfed and formula-fed, are still scarce. In addition, the growth of breastfed infants is characterized by a longer recovery of their birth weight with a slower growth and weight gain compared to formula-fed infants up to 18 months of age [55,56]. Since the WHO growth charts are based on data from healthy infants, predominantly breastfed for at least 4 months, these charts are the most appropriate option to develop a PBPK model for breastfed infants. However, these growth charts are still not entirely suitable for PBPK models for EBF infants, and the differences in the growth of EBF infants should be further studied.

The infant’s diet has a relevant impact on ontogeny and consequently, the metabolism of xenobiotics including medicines. Differences in caffeine and dextromethorphan elimination by the CYP450 enzyme superfamily were observed, with a faster maturation in formula-fed infants than in breastfed infants [61]. The knowledge of the effects of the infant’s diet medicine metabolism due to the influence on ontogeny is still limited. However, these effects are necessary to improve the predictive performance of a PBPK model specific for breastfed infants.

As mentioned, many physiological time-dependent data currently integrated in postpartum, lactation or infant PBPK models have been collected by non-systematic, but rather limited and incomplete, search strategies. This means that essential physiological data or subpopulations might be missing when building the populations in the PBPK software and predicting medicine exposure. Furthermore, these physiological data are rapidly evolving during postpartum, lactation and infancy, which emphasises the need for a high-resolution dynamic model of these stages. Additionally, a high variability in physiology in these populations is observed, next to the variability caused by non-standardized study methods in clinical studies and reviews. To truly expand our knowledge on the underlying physiological data, additional observed data from clinical studies and large datasets, e.g., meta-analyses, are needed to minimize variability [73]. To improve the translation of the dynamic physiological processes into mathematical equations within the PBPK models describing the postpartum, lactating and infant populations, systematic and thorough data collection will be necessary.

## 6. Conclusions

PBPK modelling has been embraced by academia, the pharmaceutical industry and regulatory agencies to predict medicine concentrations in special populations. However, physiological data in the postpartum, lactating and infant populations are still scattered, poorly standardized and limited. This lack of data is a limitation on the predictive performance of these PBPK tools. Currently, there is an urgent need for a systematic collection of data on the physiology of these populations, due to non-standardized methods in clinical study methods. In addition, the high population variability should be considered when incorporating maturational physiological data. The rapidly changing character of the physiological processes is one of the main challenges of incorporating these data into PBPK software. Based on systematic searches, these processes should be converted to mathematical equations of physiological time-varying data to develop and refine existing postpartum, lactation and infant PBPK models.

## Figures and Tables

**Figure 1 pharmaceutics-15-02618-f001:**
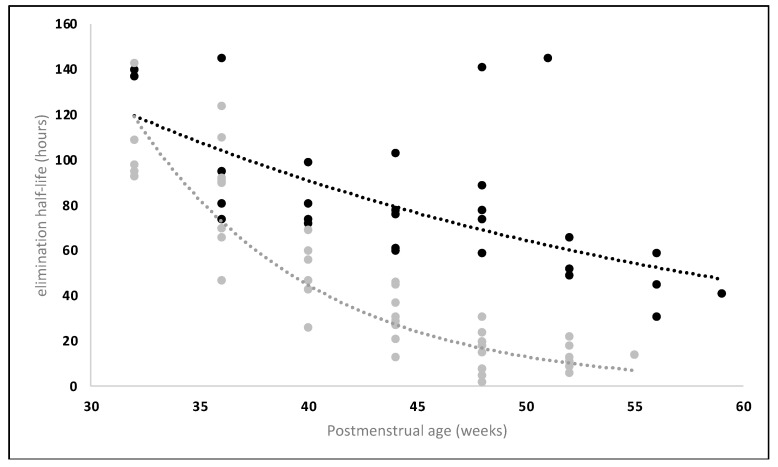
Effect of diet (human milk, black; formula, grey) on the elimination half-life of caffeine in function of postmenstrual age (30–60 weeks) and its trend lines. Adapted from Le Guennec et al. (1987) [62].

**Table 1 pharmaceutics-15-02618-t001:** Effect of infant diet (formula versus human milk) on the urinary molar ratio of caffeine (1,7-dimethyl- and 1-methyl-xanthine (1,7 MX and 1-MX)/caffeine) and dextromethorphan (3-hydroxymorphinan/dextromethrophan, 3-HM/DX), reflecting cytochrome P450 (CYP) 1A2 and 3A4 phenotypic activity, respectively, with increasing postnatal age (weeks). Adapted from Blake et al. (2006) [61].

Compound	Postnatal Age (Weeks)	Formula-Fed Infants	Breastfed Infants
	2–3	<0.01	<0.01
(1,7 MX + 1-MX)/**caffeine**	4–6	<0.01	<0.01
urinary molar ratio	8–10	0.08	<0.01
	12–15	0.48	0.04
	16–20	0.89	0.14
	24–30	3.44	0.46
	2–3	0.25	0.16
3-HM/**dextromethorphan**	4–6	0.41	0.32
urinary molar ratio	8–10	0.92	0.62
	12–15	1.22	0.73
	16–20	1.01	0.94
	24–30	1.21	0.92

## Data Availability

Data sharing is not applicable.

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
