# Peer review of "Challenges Related to Acquisition of Physiological Data for Physiologically Based Pharmacokinetic (PBPK) Models in Postpartum, Lactating Women and Breastfed Infants—A Contribution from the ConcePTION Project"

_pharmaceutics, 2023, doi:10.3390/pharmaceutics15112618_

Round 1
Reviewer 1 Report
Comments and Suggestions for Authors
The manuscript by Van Neste et al, while “aiming to illustrate the amount” of data required to inform PBPK models in postpartum, lactating, and infant populations, instead provides a cursory overview of specific data elements. In fact, even the title is misleading; while the authors discuss need for standardization, they do not make recommendations on how to acquisition of data could be more standardized. I would suggest adding such information to the Discussion section.
Overall, the manuscript feels disjointed. The authors, while calling for systematic literature reviews, only provide cursory information on examples: postpartum maternal weight retention; human milk intake and consumption by the neonate; and physiology of breastfed infants. Within these sections, the authors introduce unrelated concepts, e.g. a paragraph on the prevalence of GDM discusses hyperglycemia effects but not how GDM may be an important covariate when describing postpartum weight loss.
Several areas of the discussion session seem to repeat components of the examples (e.g. differences in maturation between breast and formula-fed infants), without further expanding on limitations in data, how these limitations could be overcome, or how standardized data collection could improve upon our knowledge.
Importantly, it is unclear how this manuscript will directly enhance PBPK modeling efforts. The last line of the abstract notes, “Following data integration as equations…”, yet few (an no novel) equations are provided in the manuscript. The manuscript appears more descriptive than quantitative, and it is unclear how the limited data elements provided could be incorporated to enhance PBPK models.
Minor comments:
I suggest additional references by other groups (e.g. Abduljali) in line 68
There are several grammatical/typographical errors. For example (not inclusive):
- Line 39. Please change “hold” to “holds”
- Lines 53-57. I suggest using e.g. instead of i.e. and removing “among others”
- Lines 95-96 appear to be missing a verb prior to population-specific changes
- Line 361: “build” should be “built”
Lines 381-382. The MPRINT Hub was established by the National Institute of Child Health and Human Development (NICHD).
Comments on the Quality of English Language
Minor grammatical errors -- see above
Reviewer 2 Report
Comments and Suggestions for Authors
The authors present an informative and generally well written review related related to physiologic change in women postpartum and during lactation and in neonates/infants that are either breastfeeding or formula fed. The main comment of this reviewer is that the title doesn't reflect the review content very closely. Perhaps a more appropriate title would be perhaps a better title would be "Challenges related to the acquisition of physiological data for PBPK models in post-partum, lactating women and breatfed infants -= A contribution from the ConcePTION project". Other comments (minor) are outlined below.
1. Line 74, change diabetes type II to type II diabetes.
2. Line 80, change similar to similarly
3. Line 138, weight and waist circumference changes post-partum are discussed. Given typical ages when pregnancy occurs, are there similar changes with ageing that are observed (i.e., in that age range, what is the year-to-year change in these values in the non-pregnant population).
4. Line 147 - not sure who "they" are. Is this normal weight? Obese? If obese perhaps you could say "from 6 weeks postpartum, obese women showed lower mean weight retention compared to normal weight women (2kg vs 4kg) but obese women had a higher fat percentage (39% vs 46%) and. higher waiste circumference (98cm vs 88cm).
***suggested text only for clarification***
5. sequence of values for obese vs normal weight (circumference, wt not lost etc.) lines 148-150 is not clear. Please ensure that this is clarified. eg. some of the ordering is confusing - perhaps (4kg in normal weight vs 2kg in overweight and obese)
6. Line 155 - respectively ?in obese women? - please clarify
7. Line 264 and vicinity - feeding moment- is this a feeding instance? feeding episode? (moment doesn't seem to describe this adequately). Please consider changing for clarity.
8. Line 272 - put parentheses after factors for clarity (as opposed to after environmental).
9. line 274 - lactation is mentioned as an important environmental factor - why isnt' this included in the factors in parentheses above?
10. line 277 - consider "evidenced" instead of supported
11. line 284 - are you describing the pbpk model here - if not - then "subsequently developed" is not appropriate - need to consider the reader here - can indicate "in a pbpk model developed by our group ..."
12. line 286 - were simplified instead of has been simplified
13. line 309 - at birth there hasn't been any feeding with breast or formula so this statement is nonsensical. At 4 months - 28 vs 28.3% is a tiny difference ...
the later part of the paragraph (on page 7) makes more sense = with at least a few tenths of a kg difference on average. For all of these comparisons, please include the p-value for these differences (given the authors assert these are significant differences).
14. line 323 "finally, the relevance of human milk co-exposure ..." would be clearer
15. line 324 "finally, the relevance of human milk co-exposure ..." would be clearer
16. line 326 Overall, the caffeine ...
17. line 328 don't need the "interestingly," just start with "This increase..."
18., table 1 label this more clearly - this reviewer assumes these are the ratios for formula fed and human milk fed neonates and infants?
19. Line 377/378 in the discussion this point about inadequacy is not supported by the review - it can always be improved but one would qualify the models with data as they become available ...
20. line 382/383 - consider clarifying to indicate that this is the mother.
21. lines 394 and 395 what impact is a few kg going to have om this - perhaps some more direct comment given the anticipated ppwr vs gwg ...and the anticipated effect postpartum on disposition more specifically
22. line 409 - "not equal to a systematic" - the authors don't specify the differences in the manuscript - and the statement surrounded by dashes does not add to the phrase.
23. lines 433 and 434 - please specify why the growth charts are not suitable for PBPK - please recapitulate why these are severely limited.
24.Line 444 clarify exactly what structured vs systematic is ... it is not clarified in the body of the manuscript (and appears earlier as well - see comment #22.
25. line 459 - the authors assert that predictive performance of the models is poor - however, it is the absence of data that limits the evaluation of the predictive performance - we don't know that the models are not predicting well ... consider clarifying this comment.
Comments on the Quality of English LanguageEnglish is quite reasonable - a few sections need editing/clarification - see comments to the authors above
Reviewer 3 Report
Comments and Suggestions for Authors
The review is a careful work, but it is heavily shifted towards physiology. I would recommend the extension of the PK part. Physiology should serve only as the background of the pharmacology (PK) in this case.
Major
1) In the first part of the review, the physiological background of maternal weight retention, human milk intake, breastfed infants, etc. were addressed in details. However, I would expect more focus on actual PK data and models available. Maybe, a separate Section could be opened using the last part of Section 4 (Table 1, Fig. 1) and including more PK models illustrated by some new Figs and/or collected in comparative Table(s).
2) The CYP450 example of Section 4 is interesting from a PK point of view. However, the explanation of Table 1 (Fig. 1) is still missing. It is mentioned that "enzyme inducers in formula (e.g., aryl hydrocarbon receptor ligands)" -- at line 350 -- may be responsible for faster metabolism (elimination) of exogenous compounds. Exactly what kind of aryl hydrocarbon-like ligands are there in the formula? Enzyme induction by - often carcinogenic - aryl hydrocarbons (or similar compounds) sounds rather weird in this context. Please, give other/more precise explanation for this important example.
For the Conclusions:
3) "Currently, there is an urgent need for data standardisation on the physiology of these populations" Please, be more explicit. Which data (examples) and what kind of standards do you mean?
4) "these processes should be converted to mathematical equations of physiological time-varying data" Please, show such equations in the previous Sections and refer back to them here.
Minor
1) I would write "cytochrome P450 (CYP450) enzyme superfamily" (line 326), as CYP450 refers to more than one enzyme.
2) Please, comment on the possible toxicity of D2O and mention the safe dose/concentration used in the "dose-to-the mother technique" . While application of D2O is obviously "non-invasive" (line 202), it has certain biological effects beyond H2O which may also complicate the analysis of the results of such investigations. See e.g. Kushner et al. an J Physiol Pharmacol. 1999 Feb;77(2):79-88 for a review.
Round 2
Reviewer 1 Report
Comments and Suggestions for Authors
Prior comments have been addressed. I have no further comments.
Reviewer 3 Report
Comments and Suggestions for Authors
-